# Five-Year Comparative Efficacy of Everolimus-Eluting vs. Resolute
Zotarolimus-Eluting Stents in Patients with Acute Coronary Syndrome Undergoing
Percutaneous Coronary Intervention

**DOI:** 10.3390/jcm10061278

**Published:** 2021-03-19

**Authors:** Endrin Koni, Wojciech Wanha, Jakub Ratajczak, Zhongheng Zhang, Przemysław Podhajski, Rita L. Musci, Giuseppe M. Sangiorgi, Maciej Kaźmierski, Antonio Buffon, Jacek Kubica, Wojciech Wojakowski, Eliano P. Navarese

**Affiliations:** 1Department of Interventional Cardiology, Santa Corona Hospital, 17027 Pietra Ligure, Italy; endrinkoni@gmail.com; 2SIRIO MEDICINE Research Network, 85094 Bydgoszcz, Poland; 3Department of Cardiology and Structural Heart Diseases, Medical University of Silesia, 40635 Katowice, Poland; wojciech.wanha@gmail.com (W.W.); kazmierski.maciej@gmail.com (M.K.); wojtek.wojakowski@gmail.com (W.W.); 4Department of Cardiology and Internal Medicine, Nicolaus Copernicus University, 87100 Bydgoszcz, Poland; ratajczak.j.m@gmail.com (J.R.); przemo.podhajski@gmail.com (P.P.); jwkubica@gmail.com (J.K.); 5Department of Health Promotion, Nicolaus Copernicus University, 87100 Bydgoszcz, Poland; 6Department of Emergency Medicine, Sir Run-Run Shaw Hospital, Zhejiang University School of Medicine, Hangzhou 310009, China; zh_zhang1984@zju.edu.cn; 7Key Laboratory of Emergency and Trauma, Ministry of Education, College of Emergency and Trauma, Hainan Medical University, Haikou 571199, China; 8Department of Biomedicine and Prevention, University of Rome Tor Vergata, 00173 Rome, Italy; muscir45@gmail.com; 9Cardiac Cath Lab, Department of Cardiology, San Gaudenzio Clinic, 28100 Novara, Italy; gsangiorgi@gmail.com; 10Institute of Cardiology, Catholic University of the Sacred Heart Rome, 00168 Rome, Italy; antonino.buffon@unicatt.it; 11Faculty of Medicine, University of Alberta, Edmonton, AB 13103, Canada

**Keywords:** everolimus-eluting stent, resolute zotarolimus-eluting stent, acute coronary syndrome

## Abstract

Among drug-eluting stents (DESs), the durable polymer everolimus-eluting stent (EES) and
resolute zotarolimus-eluting stent (R-ZES) are widely used in clinical practice and have
contributed to improve the outcomes of patients undergoing percutaneous coronary
intervention (PCI). Few studies addressed their long-term comparative performance in
patients with acute coronary syndrome (ACS). We aimed to investigate the 5 year
comparative efficacy of EES and R-ZES in ACS. We queried ACTION-ACS, a large-scale
database of ACS patients undergoing PCI. The treatment groups were analyzed using
propensity score matching. The primary endpoint was a composite of mortality, myocardial
infarction (MI), stroke, repeat PCI, and definite or probable stent thrombosis, which was
addressed at the five-year follow-up. A total of 3497 matched patients were analyzed.
Compared with R-ZES, a significant reduction in the primary endpoint at 5 years was
observed in patients treated with EES (hazard ratio (HR) [95%CI] = 0.62
[0.54–0.71], *p* < 0.001). By landmark analysis, differences
between the two devices emerged after the first year and were maintained thereafter. The
individual endpoints of mortality (HR [95%CI] = 0.70 [0.58–0.84],
*p* < 0.01), MI (HR [95%CI] = 0.55 [0.42–0.74],
*p* < 0.001), and repeat PCI (HR [95%CI] = 0.65 [0.53–0.73],
*p* < 0.001) were all significantly lower in the EES-treated patients.
Stroke risk did not differ between EES and R-ZES. In ACS, a greater long-term clinical
efficacy with EES vs. R-ZES was observed. This difference became significant after the
first year of the ACS episode and persisted thereafter.

## 1. Introduction

The advent of modern, second-generation coronary drug-eluting stents (DESs) has contributed
to improve the outcomes of patients with coronary artery disease (CAD) undergoing
percutaneous coronary intervention (PCI) by refining metallic design, polymer coatings, and
the introduction of newer antiproliferative drugs [1]. Among CAD patients, those with acute coronary syndrome
(ACS) constitute a high-risk cohort experiencing more frequent adverse clinical events
[2]. Extensive research in ACS has
mainly been conducted on outcomes at the short or medium-term follow-up [3]. However, there is a paucity of data
concerning the long-term comparison of second-generation DES in the ACS context.

Zotarolimus-eluting resolute stents (R-ZESs) (Medtronic) and everolimus-eluting stents
(EESs), such as Xience (Abbott Vascular) and Promus Element (Boston Scientific), are
second-generation DES widely used in clinical practice. Two studies have reported comparable
outcomes in the long-term between these devices [4,5]. However, they were
all-comers trials with many stable CAD patients included.

Within this framework, a relevant clinical question is whether DES-type selection in ACS
may affect long-term outcomes in these patients. We aimed to investigate the 5-year efficacy
profile of these durable polymer DES devices (R-ZES vs. EES) in ACS patients undergoing PCI,
by querying a large-scale multicenter prospective registry.

## 2. Methods

We retrospectively analyzed the ACTION-ACS, a pooled large patient ACS database involving
two interventional large-volume academic centers based in Poland: the Department of
Cardiology and Structural Heart Diseases, Medical University of Silesia, Katowice, and the
Nicolaus Copernicus University, Bydgoszcz, Poland. Information on the follow-up events was
site-reported and adjudicated by a trained physician-investigator. Data of consecutive
patients with ACS, undergoing PCI with EES (Xience, Abbott Vascular, Santa Clara, CA, USA
and Promus Element, Boston Scientific, Natick, MA, USA) or R-ZES (Resolute Integrity,
Resolute Onyx, Medtronic, Fridley, MN, USA), were collected between November 2009 and
February 2017 (Figure 1). Relevant
baseline information, procedural, and clinical outcomes at follow-up were entered into
prespecified electronic case report forms. Angioplasty and stent selection were performed
according to standard techniques at the discretion of the interventional cardiologist. All
patients were prescribed dual antiplatelet therapy (DAPT) consisting of acetylsalicylic
acid, 75 to 100 mg daily, and a P2Y12 inhibitor for at least 1 year according to
guidelines.

### 2.1. Study End Points

The primary prespecified endpoint, major adverse cardiovascular and cerebrovascular
events (MACCEs), defined as a composite of death, myocardial infarction (MI), definite or
probable stent thrombosis, repeat PCI and stroke, was addressed at the five year follow-up
[6].

Th secondary endpoints included death, repeat PCI, MI, and stroke. MI was defined
according to its fourth universal definition [7].

### 2.2. Statistical Analysis

Categorical variables are reported as n (%) and continuous variables as means ±
standard deviation. Categorical variables were compared by χ^2^ or
Fisher’s exact tests, as appropriate. Continuous data were analyzed by the
independent-samples *t*-test.

We performed an adjusted analysis based on the propensity score (PS). The PS is the
probability that each individual patient is included in the treatment group and was
estimated via logistic regression based on the available baseline covariates [8]. Potential confounders were entered into
the PS model based on known clinical relevance of associations (*p* <
0.01) observed at univariate analysis. The final variable selection was performed by a
logistic or Cox regression model with least absolute shrinkage and selection operator
(LASSO) penalty and a tuning parameter selected by cross-validation, which allows
minimizing overfitting [9].

Missing data were present in less than 5% of inspected variables. Assuming that data were
missing at random, we used polytomous logistic regression, logistic regression, and
predictive mean matching as multiple imputation techniques to fill in missing values,
using the R mice package. Matching was performed with the use of a full matching algorithm
that minimizes biases [10]. The
covariate balance was assessed by exploring the standardized mean differences between
unadjusted and adjusted populations and distribution of the PS. The standardized
differences were estimated for all the baseline covariates before and after matching to
assess the pre-match imbalance and the post-match balance. Standardized differences of
less than 10.0% for a given covariate indicate a relatively small imbalance.

The Cox proportional-hazards regression analyses were performed on the matched pairs. The
results of the Cox regression at the 5 year follow-up is presented both as an unadjusted
and adjusted hazard ratio (HR) with a 95% confidence interval (CI). Kaplan–Meier
curves were generated for the endpoints of interest and a landmark survival analysis was
conducted, setting one year as the landmark time. A *p* value < 0.05 was
considered statistically significant for all analyses. For the subgroup analyses, the p
interaction was calculated and a value of <0.10 was considered significant. The
adjusted statistical analysis was performed using R 4.0 with mice, survival and matchthem
packages.

## 3. Results

From an initial cohort of 5903 unmatched patients, a total of 3497 matched patients with
ACS were analyzed. The remainder was excluded because long-term outcomes were not complete.
All ACS patients with non-ST segment elevation myocardial infarction (NSTEMI) or segment
elevation myocardial infarction (STEMI) had acute myocardial injury with clinical evidence
of acute myocardial ischemia according to the fourth universal definition of myocardial
infarction. Among the propensity-matched patients, 3007 (86%) were NSTEMI/ unstable angina
(UA) and 490 (14%) STEMI. The clinical characteristics in the EES and R-ZES groups are
illustrated in Table 1. The average
age was 65.1 years in the EES group vs. 64.9 years in the R-ZES group. Women represented
32.3% and 33.5%, respectively. The diabetic patients represented 34.7% and 35.7%,
respectively. A radial access was predominant in both groups (88% vs. 90.6%). Multivessel
disease was present in less than 20% of the treated groups. Angina class III–IV was
observed in 73.2% and 75.4% of the treated patients, respectively. There was no acute stent
thrombosis in either group. The mean follow-up was 1686 days in the EES group and 1652 days
in the R-ZES group.

### 3.1. Primary Endpoint

Patients treated with EES stent had a significantly lower risk of MACCE at 5 years: 36.8%
vs. 46.3% (HR [95% CI] = 0.62 [0.54–0.71], *p* < 0.001, Figure 2A). By landmark analysis, the
cumulative rate difference between the two devices became statistically significant after
one year (Figure 2B). Although the
separation of the curves was observed earlier, within the first year, there was no
significant difference between EES and R-ZES: HR [95% CI] = 0.98 [0.81–1.19],
*p* = 0.88. In contrast, a significant MACCE reduction emerged after the
first year of follow-up: HR [95%CI] = 0.59 [0.48–0.73], *p* <
0.001.

### 3.2. Secondary Endpoints

#### All-Cause Mortality

Patients receiving a EES had significantly lower all-cause mortality rates compared
with those treated with a R-ZES: 12.6% vs. 17.6%, HR [95% CI] = 0.70 [0.58–0.84],
*p* < 0.01 (Figure 3).

### 3.3. Myocardial Infarction

The DES-treated patients allocated to the EES cohort experienced significantly lower
rates of myocardial infarction compared with those treated with a R-ZES: 9.8% vs. 13.8%,
HR [95% CI] = 0.55 [0.42–0.74], *p* < 0.001 (Figure 4).

### 3.4. Repeat PCI and Stroke

A significantly lower number of repeat PCI procedures were observed in patients treated
with EES vs. R-ZES: 27.8% vs. 35.6%, HR [95%CI] = 0.65 [0.53–0.73],
*p* < 0.001 (Figure 5).

No significant difference was found in stroke risk between EES and R-ZES: 4.7% vs. 5.4%:
HR [95%CI] = 0.84 [0.61–1.17], *p* = 0.32.

### 3.5. DES Performance in Pre-Specified Subgroups

Several prespecified subgroups were explored to compare EES with R-ZES. They included
sex, diabetes, dyslipidemia, Canadian Cardiovascular Society (CCS) III–IV class
angina, New York Heart Association (NYHA) III–IV, multivessel disease, left main
PCI, and previous PCI. The estimates were directionally consistent in favor of EES in
magnitude and direction without significant interactions (Figure 6).

## 4. Discussion

The main findings of this large-scale analysis of 3497 ACS propensity-matched patients
treated with EES or R-ZES drug-eluting stents followed up to five years are the following:
(1) compared with R-ZES, EES yielded a reduced risk of the primary composite endpoint of
mortality, myocardial infarction, stroke, repeated percutaneous coronary intervention, and
stent thrombosis; (2) mortality was significantly reduced in the EES-treated patients; (3) a
significant reduction in individual ischemic endpoints of MI and repeat PCI was observed;
(4) by landmark survival analysis, we identified the temporal window of greater event
reduction with EES, which occurred after one year, while no significant differences between
the EES and R-ZES emerged earlier; (5) the reduction in adverse events in the EES cohort was
consistent in all prespecified subgroups.

Newer-generation permanent polymer DESs offer numerous improvements over their
first-generation counterparts [11]. Each
of the three stent components (metallic backbone, polymer coating, and antiproliferative
drug) has undergone refinements. These features include decreased strut thickness, improved
flexibility/deliverability, enhanced polymer biocompatibility/drug elution profiles, and
superior re-endothelialization kinetics [12,13]. Available platforms
are composed of cobalt-chrome or platinum-chrome. Cobalt alloys are widely used in
new-generation DESs as they provide better radiopacity and radial strength [14].

Currently, ACS remains a high-risk setting associated with worse outcomes than stable
patients [15,16,17].
Within this framework, many factors can affect the long-term results of DES and one of those
is the polymer, which has been linked to an enhanced vascular inflammatory response [18]. The enduring inflammatory response
might lead to late stent thrombosis and restenosis owing to a combination of delayed
re-endothelialization, late-acquired incomplete apposition, neointimal hyperplasia, and
neoatherosclerosis, which in turn can impact long-term clinical outcomes [19,20,21,22,23].

In the ACS setting, vascular healing is delayed at the culprit site of the DES implantation
for an ACS episode compared with stable CAD [24]. One of the most relevant complications with the use of first-generation DES
was the high incidence of late and very late stent thrombosis (ST) concerning a bare metal
stent (BMS) [24,25,26].
Although the advent of second-generation DESs afterward has reduced the incidence of ST,
their permanent metallic scaffold prevents a complete recovery of vascular structure and
function, which may have implications on very late stent failure [1,27]. In
this regard, DES components, such as the drug-eluting polymer and their biocompatibility,
are key determinants for long-term device success. ACS is associated with higher risk of
adverse outcomes owing to the proinflammatory state and higher atherosclerotic burden [16]. Besides pharmacological and technology
advancement, a persistent risk of adverse events remains over the 1 year period after an ACS
event [15,28,29].
Thus, a DES design aimed to provide a more biocompatible polymer and a better polymer/drug
elution balance remains a pivotal goal in ACS [21,30].

In both EES and R-ZES platforms, drug release is completed within 180 days [31]. Thus, it is unlikely that the observed
long-term stent differences between these two DES in our study were drug-related.
Conversely, the associated durable polymer could play a key role in the modulation of DES
performance over the long-term. Accordingly, R-ZESs incorporate BioLinx, which is a mixture
of hydrophobic C10, hydrophilic C19, and polyvinyl pyrrolidone polymers. Polymer-orientation
results in a hydrophilic surface and hydrophobic core with enhanced biocompatibility [32]. At variance with R-ZESs, the EES
polymer consists of a bilayer copolymer (vinylidene fluoride-co-hexafluoropropylene)
(PVDF-HFP), which is highly fluorinated and referred to as a fluoropolymer [33]. The two components of the polymer
demonstrate high biocompatibility. Fluorinated copolymers, such as vinylidene
fluoride–hexafluoropropylene copolymer, reduce platelet adhesion and thrombus
formation [34]. These features are
likely related to a high retention of albumin, which in turn passivates the stent surface
and prevents fibrinogen binding [35].
Fluoropolymer coated with everolimus stent platforms also induces a lower inflammatory
response, even compared with a BMS [36].

These polymer characteristics could be implicated in greater long-term ischemia reduction
and may offer improved survival in patients treated with an EES platform compared with
R-ZES, as noted in our study. Although no substantial differences between these two devices
has emerged in a stable setting, our findings of long-term clinical improvement with EES vs.
R-ZES in ACS are in agreement with other studies that demonstrated an incremental benefit
with new-generation DES in relation to ACS severity [37].

The R-ZES and EES are the most-used durable polymer platforms among DES. The RESOLUTE US, a
prospective observational study of resolute [38] and SPIRIT FIRST [39], a
trial comparing Xience with a bare metal stent, demonstrated a good long-term outcome for
both devices. Subsequently, four studies conducted a short-term (1 year) head-to-head
comparison of R-ZES with EES [40,41,42,43].
They enrolled mainly stable CAD patients with only one of them including STEMI patients.
Thus, a paucity of comparative data between EES and R-ZES is available in the ACS field. The
resolute all-comers trial compared long-term results of R-ZES with Xience EES [44]. Among the included patients in the
trial, only 34% had an MI. Resolute and Xience stents yielded a similar efficacy and safety.
In the subgroup analysis of patients with an acute MI, which included 662 patients only,
there was a numerical nonsignificant benefit in favor of EES. The DUTCH PEERS randomized
study evaluated the 5 year outcomes of Resolute Integrity ZES and Promus Element EES [4]. Approximately 42% of included patients
were stable CAD. Resolute Integrity and Promus Element showed similar efficacy. However, in
this study, the comparative performance between devices was not addressed in the pure ACS
setting.

These studies were underpowered to conclude on individual endpoints in ACS patients.

One study evaluated very late (1–5 years) pathological response to EES (EES; Abbott
Vascular) and BMS (Multilink Vision; Abbott Vascular). A lower inflammatory response of the
EES was observed concerning the BMS platform [45]. The pathology findings translated into better clinical outcomes in the
EXAMINATION trial that included 1498 patients with STEMI [46]. At 1 year, the composite endpoint (all-cause death, MI,
or any revascularization) did not differ between the two groups (11.9% vs. 14.2%,
*p* = 0.19). Notably, at the 5 year follow-up, there were significantly
lower rates of the composite endpoint in the EES treatment group compared with the BMS (21%
vs. 26%, *p* = 0.03).

To the best of our knowledge, this is one of the largest-scale analyses of
propensity-matched patients at long-term after ACS. We observed at 5-year follow-up greater
clinical benefits with EES in comparison to R-ZES. This difference became significant after
the first year of the index event and persisted thereafter. Further randomized and
adequately powered research is warranted to confirm the relative long-term efficacy of EES
vs. R-ZES in these high-risk patients.

## 5. Limitations

This was an observational study and, unlike a randomized trial, does not account for all
confounding variables. The use of a propensity score generated a balanced population,
mitigating confounding. However, residual confounding could not be fully excluded.
Consistency of the results in all prespecified subgroups supports the reliability of the
findings. Propensity matching generated balanced groups in terms of their baseline
characteristics. Less complete information was available with precise timing at the
follow-up for stent thrombosis, which made it impossible to explore it as an individual
endpoint. Most patients included in the registry with available 5 year data were NSTEMI or
UA, while a lower fraction had a STEMI. We did not include ACS type in the propensity
matching procedure since UA and NSTEMI are part of the continuum of ACS, which also includes
STEMI. All ACS patients with NSTEMI or STEMI had acute myocardial injury with clinical
evidence of acute myocardial ischemia according to the fourth universal definition of MI.
The inclusion of ACS type in the propensity analysis would have led to a greater loss of
statical power by reducing the final sample size. In some instances, multiple stents were
implanted; however, they belonged to the same class, thereby limiting potential confounding
arising from different stents.

Because a significant difference between the two devices was observed beyond the first year
only, it is unlikely that any periprocedural factor occurring during the invasive procedure
between treatment groups could have influenced long-term outcomes. However, given the
observational design, these findings of greater efficacy of EES vs. R-ZES need further
validation in future powered randomized trials.

## 6. Conclusions

In this large-scale analysis performed in ACS patients, treatment with EES resulted in
better long-term outcomes compared with R-ZES. The benefit was observed after the first year
and persisted thereafter. These findings suggest that in patients with ACS, EESs provide a
greater 5-year clinical efficacy in comparison with R-ZESs. Further adequately powered
investigations are needed to address the long-term efficacy of these devices in ACS.

## Figures and Tables

**Figure 1 jcm-10-01278-f001:**
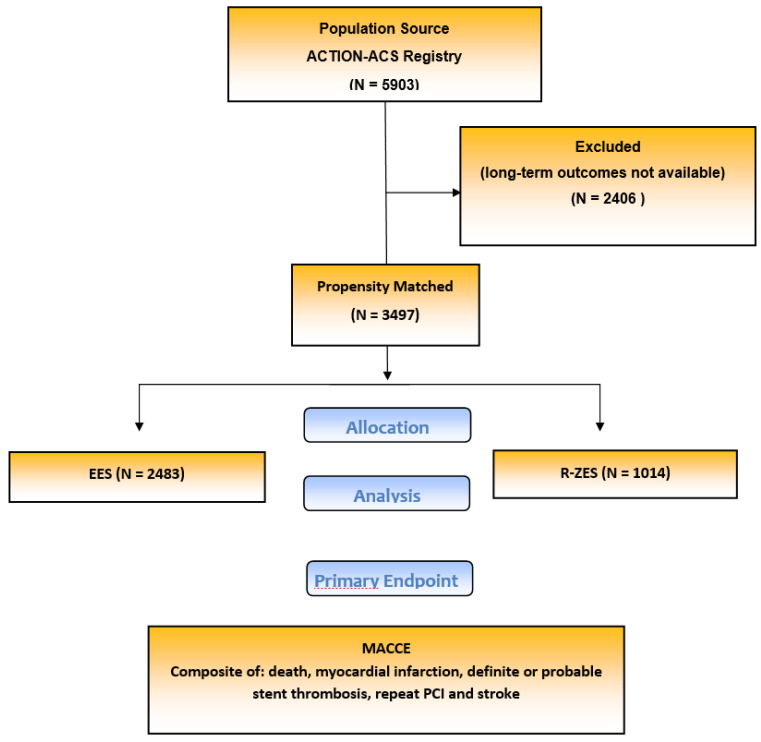
Study flow chart. EES, everolimus-eluting stent; R-ZES, resolute zotarolimus-eluting
stent; MACCE, major adverse cardiovascular and cerebrovascular events; PCI, percutaneous
coronary intervention.

**Figure 2 jcm-10-01278-f002:**
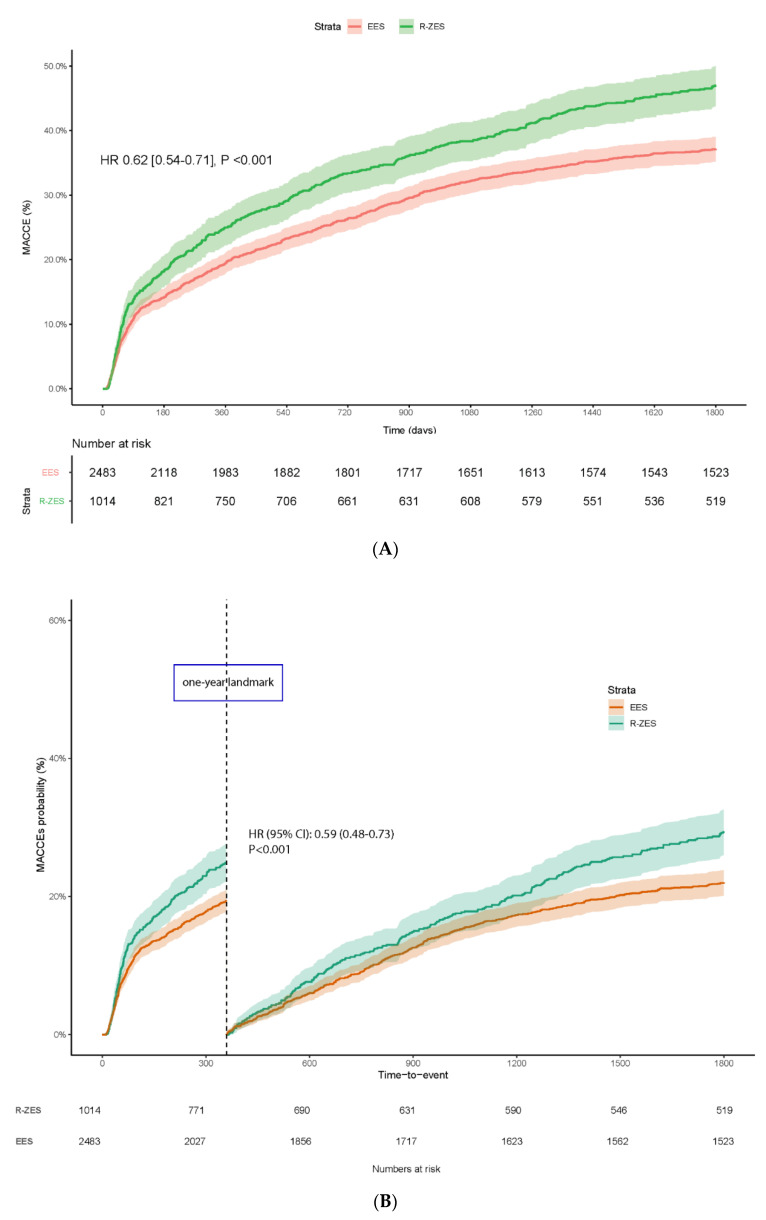
(**A**) Kaplan–Meier graph of the cumulative incidence of MACCE.
(**B**) The time-to-event landmark analysis showing event curve divergence
that became statistically significant after the one year landmark point.

**Figure 3 jcm-10-01278-f003:**
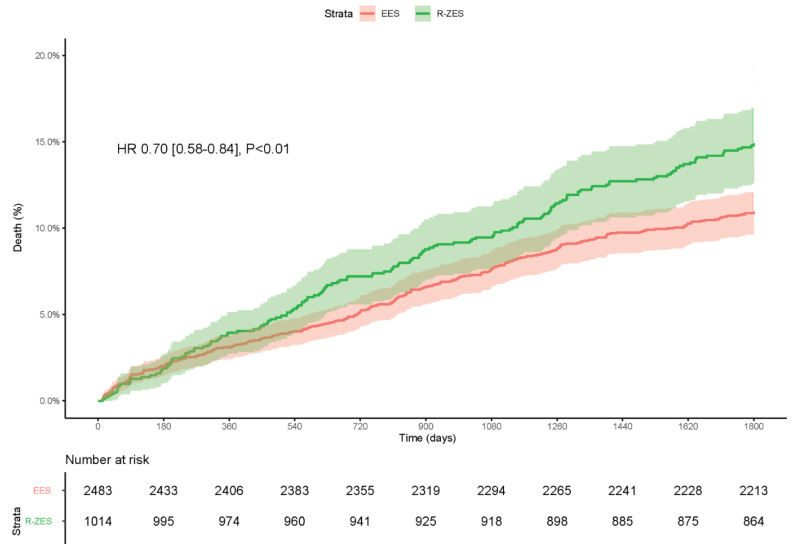
Kaplan–Meier graph of the cumulative incidence of mortality.

**Figure 4 jcm-10-01278-f004:**
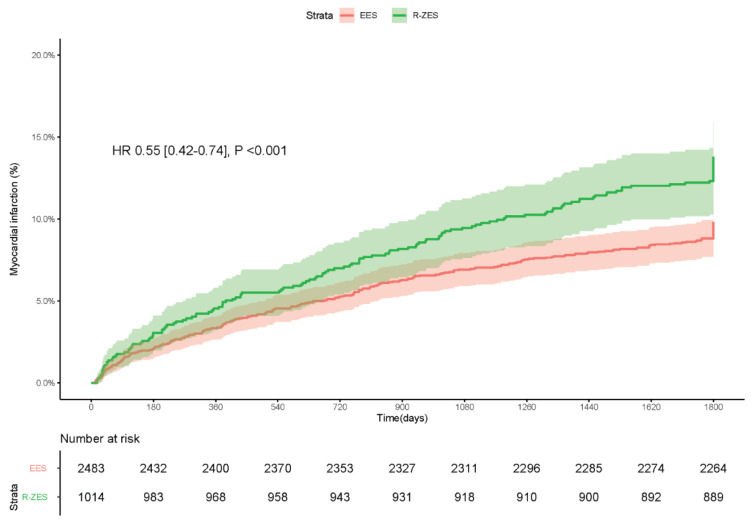
Kaplan–Meier graph of the cumulative incidence of myocardial infarction.

**Figure 5 jcm-10-01278-f005:**
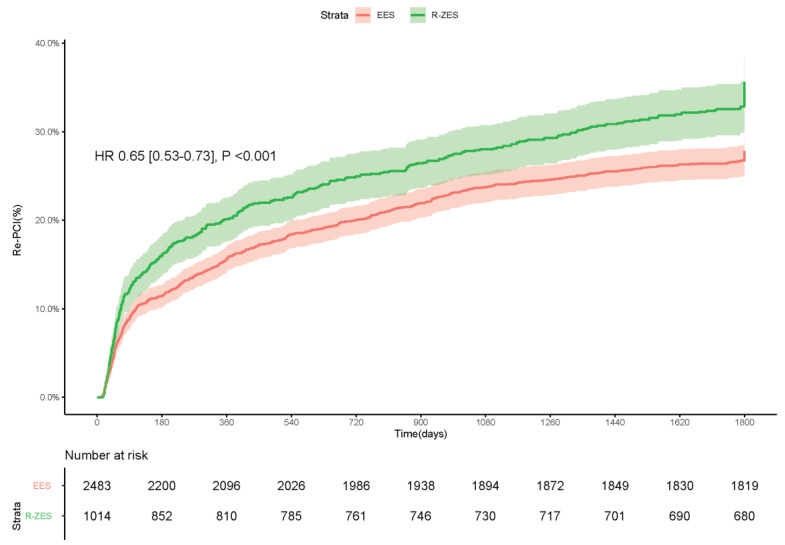
Kaplan–Meier graph of the cumulative incidence of repeat PCI.

**Figure 6 jcm-10-01278-f006:**
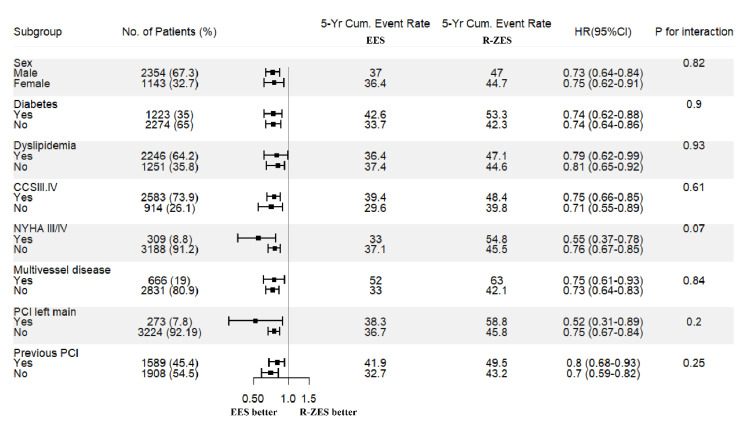
Forest plot analysis of the prespecified subgroups. CCS, Canadian Cardiovascular
Society Grading Angina; NYHA, New York Heart Association heart failure classification;
PCI, percutaneous coronary intervention.

**Table 1 jcm-10-01278-t001:** The baseline characteristics after propensity matching of ACS patients treated with EES
vs. R-ZES.

Patient Characteristics	*EES*	*R-ZES*	*p* Value
	(n = 2483)	(n = 1014)	
Women, n (%)	803 (32.3)	340 (33.5)	0.7
Age, mean ± SD	65.16 (10.3)	64.99 (10.2)	0.4
Diabetes, n (%)	861 (34.7)	362 (35.7)	0.5
CCSIII.IV, n (%)	1818 (73.2)	765 (75.4)	0.1
Hypertension, n (%)	2131 (85.8)	881 (86.9)	0.4
BMI, n (%)	28.44 (4.4)	28.16 (4.1)	0.2
Dyslipidemia, n (%)	1575 (63.4)	671 (66.2)	0.1
Anemia, n (%)	388 (15.6)	153 (15.1)	0.6
NYHA, n (%)			0.3
NYHA class II	295 (11.9)	100 (9.9)	
NYHA class III	174 (7)	64 (6.3)	
NYHA class IV	51 (2.1)	20 (2.0)	
Radial access, n (%)	2185 (88.0)	919 (90.6)	0.3
MVD, n (%)	474 (19.1)	192 (18.9)	0.9
Bifurcation, n (%)	92 (3.7)	40 (3.9)	0.7
**Adjunctive therapies**			
IABP, n (%)	25 (1.0)	9 (0.9)	0.7
Thrombectomy, n (%)	52 (2.0)	19 (1.8)	0.2
GP IIB/IIIa inh, n(%)	151 (6.1)	52 (5.1)	0.4
**Culprit vessel PCI**			
PCI Cx, n (%)	596 (24.0)	224 (22.1)	0.2
PCI LAD, n (%)	1135 (45.7)	469 (46.3)	0.7
PCI RCA, n (%)	654 (26.3)	258 (25.4)	0.5
PCI LM, n (%)	107 (4.3)	34 (3.4)	0.1
Residual stenosis, n (%)	13 (0.5)	4 (0.4)	0.4

SD = standard deviation; BMI = body mass index; LAD = left anterior descending; Cx =
circumflex artery; GP IIB/IIIa inh = glycoprotein IIb/IIIa inhibitor; LM = left main;
RCA = right coronary artery; IABP = intraortic balloon pump counterpulsation; MVD =
multivessel disease; NYHA = New York Heart Association heart failure classification;
CCS = Canadian Cardiovascular Society Grading Angina.

## Data Availability

Upon reasonable request after all authors’ signed approval.

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

myocardial infarction (EXAMINATION): 1 year results of a randomised controlled
trial. Lancet.

