# Peer review of "Five-Year Comparative Efficacy of Everolimus-Eluting vs. Resolute Zotarolimus-Eluting Stents in Patients with Acute Coronary Syndrome Undergoing Percutaneous Coronary Intervention"

_jcm, 2021, doi:10.3390/jcm10061278_

Round 1

Reviewer 1 Report

The authors give a good review and analysis of an older and newer DES. This topic is very important as in-stent re-stenosis can lead to new symptoms and even MI.

The grammar and English was very good and the paper is well written and easy to follow.  Importantly, the paper is concise. The Kaplan Meier graphs were especially helpful.

As a neurointerventionalist, this creates interest about the possible use of these stents for neurovascular disease.

Author Response

We thank the Reviewer for his positive comments. We appreciate the value given to this relevant manuscript for the cardiology community. We are also glad if this work could contribute even to other fields.

Reviewer 2 Report

In their study, the authors present a retrospective study comparing the long-term efficacy of two durable 2nd generation DES. While the topic is certainly of great interest to the readership, the not good grammar, spelling and lack of care in formatting and revising the text makes it very hard to read. The whole manuscript needs extensive revision in language and grammar. I recommend using an English language service or, for example, the online platform deepl.com.

In its present form and due to the lack of care in the preparation and accuracy in the description of the statistical methods, I recommend that the manuscript not be published in this form.

Page 2, line 33: Please introduce all abbreviations within the abstract! (PCI, CAD, ACS, HR, MI)

Page 2, line 35: “Second-generation drug-eluting stents (DES) have contributed to improving the outcomes in comparison to first-generation DES.” and “… high-risk …” and “one-year”

Page 2, line 50: “In a time-to-event landmark analysis, we observed this difference 1 year after the ACS event.”

Page2, line 55: “In ACS, the Xience stent performed better than the Resolute stent, reducing the primary endpoint, mortality and re-PCI during a 5-year follow-up. The benefit became significant after the first year of the index event and persisted on.”

Page 3, line 74: Please provide name, registry number etc.

Page 3, line 87: Please pay attention to the formatting!

Page3, line 85: Please introduce abbreviations appropriately.

Page 4, line 115: Please indicate for which co-variables the COX regression model was corrected. Have you considered conducting Kaplan-Meyer Survival estimates.

Page 4, line 120: Please cite R Software and packages properly!

Page 5, table 1: Please indicate all abbreviations at the end of the table.

Page 7, Fig.1: Please describe estimation of Kaplan-Mayer Survival in the methods part.

Author Response

In their study, the authors present a retrospective study comparing the long-term efficacy of two durable 2nd generation DES. While the topic is certainly of great interest to the readership, the not good grammar, spelling and lack of care in formatting and revising the text makes it very hard to read. The whole manuscript needs extensive revision in language and grammar. I recommend using an English language service or, for example, the online platform deepl.com.

REPLY. We remain surprised by the reviewer comments. Aside from typos or very minor spelling mistakes that have now been corrected, the overall English grammar is sound as confirmed by the first reviewer. The text of the manuscript underwent an extensive language revision, after which we believe that the study is well presented. 

In its present form and due to the lack of care in the preparation and accuracy in the description of the statistical methods, I recommend that the manuscript not be published in this form.

REPLY- We disagree with the Reviewer comments on this point. We are seriously concerned that the reviewer did not have a careful read of the statistical method paragraph. Employed statistical methods are indeed very robust (propensity score analysis to generate a quasi-randomized design,  covariate balance control, least absolute shrinkage-LASSO, cox regression, Kaplan Meier curve with number at risk, landmark analyses, subgroup analyses) a strength of this large-scale analysis which are very well presented. The senior author of this manuscript is an expert in the field of  methodology, evidence-based medicine a and editor of the Cochrane, London, United Kingdom.

 In detail in the methods we write:  An adjusted analysis based on propensity score (PS) was performed. PS is the probability that each individual patient is included in the treatment group and was estimated via logistic regression  based on the available baseline covariates.2 Potential confounders were entered into the PS model on the basis of known clinical relevance  of associations (P<.10) observed at univariate analysis; final variable selection was performed by a logistic or Cox regression model with LASSO (Least Absolute Shrinkage and Selection Operator) penalty and a tuning parameter selected by cross-validation, which allows to minimize overfitting.3

Missing data were present in less than 5% of inspected variables. Assuming that data were missing at random we used polytomous logistic regression, logistic regression and predictive mean matching as multiple imputation techniques to fill in missing values, using the R mice package.

Matching was performed with the use of a nearest neighbor matching algorithm with replacement which minimizes biases4  and a caliper width equal to 0.2 of the standard deviation of the logit of the propensity score, as this value was associated with minimized  mean squared error of the estimated treatment effect.2Covariate balancement was assessed exploring the standardized mean differences between unadjusted and adjusted populations and distribution of the PS.   Standardized differences were estimated for all the baseline covariates before and after matching to assess pre-match imbalance and post-match balance. Standardized differences of less than 10.0% for a given covariate indicate a relatively small imbalance.

Cox proportional- hazards regression analyses were performed on the matched pairs. The results of the Cox regression at 5 year- follow-up are presented both as unadjusted and adjusted hazard ratio (HR) with 95% confidence interval(CI).

Page 2, line 33: Please introduce all abbreviations within the abstract! (PCI, CAD, ACS, HR, MI)

REPLY. Abstracts abbreviations have been introduced.

Page 2, line 35: “Second-generation drug-eluting stents (DES) have contributed to improving the outcomes in comparison to first-generation DES.” and “… high-risk …” and “one-year”

Page 2, line 50: “In a time-to-event landmark analysis, we observed this difference 1 year after the ACS event.”

Page2, line 55: “In ACS, the Xience stent performed better than the Resolute stent, reducing the primary endpoint, mortality and re-PCI during a 5-year follow-up. The benefit became significant after the first year of the index event and persisted on.”

Reply. We have now rephrased these sentences in a more concise and clear way

Page 3, line 74: Please provide name, registry number etc.

REPLY The database has been locally approved as  a retrospective analysis which we clarified in the manuscript. We also report the associated ID which is ACTION-ACS registry. We also provided in the revision version a Flowchart to make the data streamline more clear.

Page 3, line 87: Please pay attention to the formatting!

REPLY as stated above, an extensive text and formatting revision has now been conducted.

Page3, line 85: Please introduce abbreviations appropriately.

REPLY- We have now introduced abbreviations properly.

Page 4, line 115: Please indicate for which co-variables the COX regression model was corrected. Have you considered conducting Kaplan-Meyer Survival estimates.

REPLY. We are very surprised by this Reviewer statement. We have conducted KM curves which are currently displayed in the figures 2A, 2B, 3, 4 and 5!!!

Page 4, line 120: Please cite R Software and packages properly!

REPLY. We cited properly the R software and its related packages. We have clarified  this point in the revision.

Page 5, table 1: Please indicate all abbreviations at the end of the table.

REPLY. We have done so. 

Page 7, Fig.1: Please describe estimation of Kaplan-Mayer Survival in the methods part.

REPLY, We have further clarified in the methods that we have conducted both Kaplan Meier curves and landmark survival analysis.

Reviewer 3 Report

Navarese E. P et al provided a long-term comparison between two second-generation DES Xience and Resolute) in ACS patients. Xience stent seems to reduce MACCE after one-year.

Authors should be congratulated for their work and a valuable cohort (n=3673 patients). Furthermore, long-term follow-up are interesting and should be promoted.

Major comments

-The article is not well-written (too long, not easy to read, too many abbreviations: even in the tittle, no space before (), etc).

-In the secondary endpoints, it would be useful to detail all items of MACCE. Furthermore, “hospitalizations” means all hospitalizations or only for cardiac reasons?

-Results are surprising: there is a huge difference between the two DES, even on mortality, that is not explained in discussion

-There are many excluded patients: 2230 (38%). Explain why?

- Among ACS, there are only 9% of STEMI: this is not a reflect of reality (about 50% STEMi, 50% NSTEMI). Furthermore, only 20% of patients have multivessel disease. This cohort is not a high-risk cohort as the authors said in introduction (line 67).

-Statistical analysis paragraph is very long and poorly understandable

-Discussion paragraph is very long also.

-The difference between Xience and Resolute on MACCE is significative afer one year: why?

Minor comment

-Please define residual stenosis?

Author Response

Navarese E. P et al provided a long-term comparison between two second-generation DES Xience and Resolute) in ACS patients. Xience stent seems to reduce MACCE after one-year.

Authors should be congratulated for their work and a valuable cohort (n=3673 patients). Furthermore, long-term follow-up are interesting and should be promoted.

REPLY. We thank the Reviewer for his/her appreciation of our work.

Major comments

-The article is not well-written (too long, not easy to read, too many abbreviations: even in the tittle, no space before (), etc).

REPLY. We have shortened and condensed the manuscript, making it more easily readable. We have now changed the title adopting a format without abbreviations and cited the specific DES- eluted drug for  clarity: “Five-year Comparative Efficacy of Everolimus-eluting vs Resolute-Zotarolimus-Eluting Stents (EES) in Patients with Acute coronary Syndrome Undergoing Percutaneous Coronary Intervention”.  We have clarified that in the group of EES were included Xience (Abbott Vascular) and Promus (Boston Scientific) stents, and in the other group Resolute Integrity/Onyx (Medtronic).

-In the secondary endpoints, it would be useful to detail all items of MACCE. Furthermore, “hospitalizations” means all hospitalizations or only for cardiac reasons?

REPLY. We have already clarified the definition of the MACCE endpoint both in abstract and in the manuscript (outcome section). The hospitalizations were doe to heart failure and we have now clarified this in the revised version.

-Results are surprising: there is a huge difference between the two DES, even on mortality, that is not explained in discussion

REPLY. We have provided  extensive discussion of the findings and potential explantions for the observed differences. We have now streamlined the discussion paragraph and made it more focused.

-There are many excluded patients: 2230 (38%). Explain why?

REPLY.  Patients were not excluded from the registry, but from propensity score analysis. Propensity score method is a very robust and established methods to account for confounding in observational studies and widely implemented in cardiovascular literature. This methodology allows to generate propensity-matched population in terms of their baseline characteristics mimicking a randomized trial. The propensity quasi-randomized design  ultimately allows to account for heterogeneity and reverse causality that may occur in observational studies. Therefore is an excellent method to provide reliable final estimates. For simplicity we have provided a study Flowchart in the revised manuscript.

- Among ACS, there are only 9% of STEMI: this is not a reflect of reality (about 50% STEMi, 50% NSTEMI). Furthermore, only 20% of patients have multivessel disease. This cohort is not a high-risk cohort as the authors said in introduction (line 67).

REPLY.  We have re-assessed all ACS patients and the percentage is higher (14%)as now stated in the revised manuscript. We state in the limitations that the majority of patients included in the registry with available 5-year data were NSTEMI or UA, while a lower fraction had a STEMI. On the other hand, All ACS patients with NSTEMI or STEMI had acute myocardial injury with clinical evidence of acute myocardial ischemia according to the fourth universal definition of MI.

-Statistical analysis paragraph is very long and poorly understandable

REPLY. The applied statistical analysis method is robust and very refined. Several methods including propensity score, hazard ratio, cox regression, Kaplan Meier curves and landmark analyses make these findings robust, 

-Discussion paragraph is very long also.

REPLY. As stated above, we have now streamlined and condensed  the discussion paragraph.

-The difference between Xience and Resolute on MACCE is significative afer one year: why?

REPLY. We have extensively explained the potential reason in discussion. A more biocompatible polymer pertaining to the EES Xience DES could have translated into the improved outcomes over time observed with Xience vs Resolute. This result is in directional agreement with findings from  other smaller studies cited in discussion. Suttle difference was observed even during the first year but not significant. Probably the therapy of the first year has mitigated the difference.

Minor comment

-Please define residual stenosis?

REPLY Residual stenosis is the stenosis which persists after PCI.

Reviewer 4 Report

The objective of this work is to compare the clinical outcomes of everolimus-eluting stent (EES) versus  zotarolimus –eluting stent  (ZES),                           the mainstay second generation DES stent in an ACS large-scale database using a propensity score matching over a five years follow- up. The authors found a significant reduction of the composite  primary endpoint in the EES group mainly explain by the difference in the poplymer plateform and not by drug related.

This topic is important  but  I disagree with their  conclusions

I have several comments and majors revisions.

1) In the “limitations of the study”, the use of propensity score doesn’t generate randomized study. This sentence must be rewritten with more moderation  

2)The primary endpoint included hospitalizations is not describied. What kind of hospitalisation ? No description in the methods and no details  in the results ( stroke CHF others )

3)There is no flow chart ? How many patients was excluded ? Why ? the reasons ?

4) What is the mean follow-up in each group ?

5) In table 1 how many patient in EES group ? in ZES group ?

6) Radial access is different in each group ? why ? can it explain the difference in the 2 groups ? explain in the discussion ?

7) The authors must describied the population in two tables

        a)First demographic data and reason for admission

        b) Second angiographic and revascularization characteristic

8) Mortality must be divided en total mortality and cardiovascular mortality in each group with two different curves.

9) Repeat PCI must be describied as stent thrombosis (ARC definition) emergent PCI, ischaemia driven PCI in each group

10) In each group, what kind of treatment ? what kind of platelet agents ? how long was the DAPT  ( mean, mediane  and extreme) ? These results must be explain in the discussion.

Author Response

Reviewer 4

The objective of this work is to compare the clinical outcomes of everolimus-eluting stent (EES) versus  zotarolimus –eluting stent  (ZES), the mainstay second generation DES stent in an ACS large-scale database using a propensity score matching over a five years follow- up. The authors found a significant reduction of the composite  primary endpoint in the EES group mainly explain by the difference in the poplymer plateform and not by drug related.

This topic is important  but  I disagree with their  conclusions

I have several comments and majors revisions.

1) In the “limitations of the study”, the use of propensity score doesn’t generate randomized study. This sentence must be rewritten with more moderation  

REPLY. We have now extended and revised the limitations section. We never stated that this analysis method generated a randomized study but a “quasi randomized study design” as it allows group balance. We removed the term quasi-randomized for misunderstanding . However, this terminology is extensively  adopted in the statistical method section of studies published in top-tier journals and by eminent experts in the fields. Using Propensity Scores in Quasi-Experimental Designs By: William M. Holmes- DOI: https://dx.doi.org/10.4135/9781452270098

2)The primary endpoint included hospitalizations is not describied. What kind of hospitalisation ? No description in the methods and no details  in the results ( stroke CHF others )

REPLY.  We have clarified now that hospitalizations were doe to heart failure (HF). We have stated both in the abstract and in the endpoint section of the manuscript the definition of the primary endpoint which is a composite of composite of mortality, myocardial infarction, stroke,  hospitalizations, definite or probable stent thrombosis, and repeat PCI. However, some data and precise timing at follow-up for heart failure hospitalizations and stent thrombosis were missing, which  made not possible to explore them as individual endpoints.

3)There is no flow chart ? How many patients was excluded ? Why ? the reasons ?

REPLY. We have now provided a Flowchart to help the reader synthesize the data. In an initial cohort of 5903 unmatched patients, the propensity score matching procedure generated a cohort of 3497 matched patients, balanced in their baseline characteristics that entered into the analysis.

4) What is the mean follow-up in each group ?

REPLY. All patients were evaluated at 5 year follow-up

5) In table 1 how many patient in EES group ? in ZES group ?

REPLY. We have now added this information in table 1.

6) Radial access is different in each group ? why ? can it explain the difference in the 2 groups ? explain in the discussion ?

REPLY. Radial access was not different since we generated a propensity score matched population.

7) The authors must described the population in two tables

        a)First demographic data and reason for admission

  1.       b) Second angiographic and revascularization characteristic

REPLY. We find dispersive to present separately the angiographic and demographic data which should be presented as they are; in this way the reader can fully capture all characteristics of the matched patients.

8) Mortality must be divided in total mortality and cardiovascular mortality in each group with two different curves.

REPLY. This analysis cannot be done. The primary prespecified  endpoint is all-cause mortality which is the same endpoint shared by  many multiple registries, meta-analysis and trials covering the drug-eluting stent subject.

9) Repeat PCI must be described as stent thrombosis (ARC definition) emergent PCI, ischaemia driven PCI in each group

REPLY.  Repeat PCI is a prespecified component primary endpoint which cannot and should not be misclassified as stent thrombosis that is a distinct endpoint. We encourage the reviewer to consult the extensive literature on the topic.

10) In each group, what kind of treatment ? what kind of platelet agents ? how long was the DAPT  ( mean, mediane  and extreme) ? These results must be explain in the discussion.

REPLY.  Duration of Dual antiplatelet therapy was one-year as per guideline recommendations in ACS. We already provided this information in the manuscript. 

Round 2

Reviewer 2 Report

Thank you for your reply. I have carefully considered your concerns and have no further comments about the statistical methods. However, there are still a few issues concerning the literature cited:

Major comments:

Page 4: The introduction part comes very short. There might be no literature adresseing xou concrete point, however, 5-year follow up comparing e.g.  XIENCE stents and bare metal stents (SPIRIT FIRST trial) and the RESOLUTE US clinical trial (2020) are available and the results are in my opinion worth to be mentioned. Also the TWENTE study (Real-World Endeavor Resolute vs Xience V Drug-Eluting Stent Study in Twente) should be mentioned and discussed: “Five-Year Outcome After Implantation of Zotarolimus- and Everolimus-Eluting Stents in Randomized Trial Participants and Nonenrolled Eligible Patients” Birgelen et al. JAMA Cardiology. 2017 and JACC Cardiovasc Interv. 2018 Mar 12;11(5):462-469. There is also literature available regarding a 2-year follow up (Unrestricted randomised use of two new generation drug-eluting coronary stents: 2-year patient-related versus stent-related outcomes from the RESOLUTE All Comers trial; 10.1016/S0140-6736(11)60395-4). Please discuss.

Minor comments:

Page 2, line 36: It should be “Resolute zotarolimus-eluting stent (R-ZES) or just ”zotarolimus-eluting stent (ZES). In your first version you named the Xience everolimus-eluting stent and Resolute zotarolimus-eluting stent. Now you write durable polymer everolimus-eluting stent. In this case you should also write zotarolimus-eluting stent (ZES) instead of Resolute zotarolimus-eluting (R-ZES). This should be this should be consistent throughout the text.

Page 2, line 48: A total of 3497 …

Page 2, line 52: missing space

Page 2, line 59: The keyword should be zotarolimus-eluting stent.

Page 3, line 75: Please use uniform abbreviations: R-ZES – Resolute zotarolimus-eluting stent or ZES zotarolimus-eluting stent

Page 4, line 89: …polymer DESs devices…

Page 4, line 95: There seems to be one space to many.

Page 4, line 102: it should be of instead of in

Page 4, line 106: missing space ...infarction(MI)....

Page 5, line 117: there seems to be a 0 missing. please correct: (P<.10)

Page 5, line 128: There seems to be one space to many.

Page 5, line 136/137: please write consistently ; There seems to be one space to many.

Page 14, line 203: ...with EES or R-ZES drug-eluting stents followed up to five years...

Author Response

Reviewer 2

Thank you for your reply. I have carefully considered your concerns and have no further comments about the statistical methods. However, there are still a few issues concerning the literature cited:

We thank the reviewer for detailed evaluation of our paper which is now improved substantially. We tried to address all your comments.

Major comments:

Page 4: The introduction part comes very short. There might be no literature addressing you concrete point, however, 5-year follow up comparing e.g.  XIENCE stents and bare metal stents (SPIRIT FIRST trial) and the RESOLUTE US clinical trial (2020) are available and the results are in my opinion worth to be mentioned. Also the TWENTE study (Real-World Endeavor Resolute vs Xience V Drug-Eluting Stent Study in Twente) should be mentioned and discussed: “Five-Year Outcome After Implantation of Zotarolimus- and Everolimus-Eluting Stents in Randomized Trial Participants and Nonenrolled Eligible Patients” Birgelen et al. JAMA Cardiology. 2017 and JACC Cardiovasc Interv. 2018 Mar 12;11(5):462-469. There is also literature available regarding a 2-year follow up (Unrestricted randomized use of two new generation drug-eluting coronary stents: 2-year patient-related versus stent-related outcomes from the RESOLUTE All Comers trial; 10.1016/S0140-6736(11)60395-4). Please discuss.

REPLY – Thank you for your suggestions.  Accordingly, we expanded the background  citing DUTCH PEERS trial and “Five-Year Outcome After Implantation of Zotarolimus- and Everolimus-Eluting Stents in Randomized Trial Participants and Nonenrolled Eligible Patients” Birgelen et al. JAMA Cardiology. 2017 trial.

We have now referenced RESOLUTE US and SPIRIT FIRST studies in the Discussion section   when commenting on long term outcomes of Resolute and Xience stents. RESOLUTE All Comers trial was  discussed subsequently.

Minor comments:

Page 2, line 36: It should be “Resolute zotarolimus-eluting stent (R-ZES) or just ”zotarolimus-eluting stent (ZES). In your first version you named the Xience everolimus-eluting stent and Resolute zotarolimus-eluting stent. Now you write durable polymer everolimus-eluting stent. In this case you should also write zotarolimus-eluting stent (ZES) instead of Resolute zotarolimus-eluting (R-ZES). This should be this should be consistent throughout the text.

Page 2, line 48: A total of 3497 …

REPLY – corrected as suggested

Page 2, line 52: missing space

REPLY - corrected

Page 2, line 59: The keyword should be zotarolimus-eluting stent.

REPLY – While we theoretically agree that all are zotarolimus-eluting stents, it would be incorrect to broadly refer to them as ZES since  such terminology would encompass also the Endeavor stent which is the older ZES counterpart  purposely excluded from this analysis as a potential confounder. Conversely, R-ZES is the most appropriate acronym when referring to this group which incorporated the most recent devices iterations  than the Endeavor stent. Of note,   the same terminology  on the comparative use of EES vs R-ZES has been widely adopted in most recent analyses (cfr Safety and efficacy of resolute zotarolimus-eluting stents(R-ZES) compared with everolimus-eluting stents(EES): a meta-analysis.Circ Cardiovasc Interv. 2015 Apr;8(4):e002223.

Page 3, line 75: Please use uniform abbreviations: R-ZES – Resolute zotarolimus-eluting stent or ZES zotarolimus-eluting stent

REPLY – we have uniformed with R-ZES throughout the text.

Page 4, line 89: …polymer DESs devices…

REPLY – corrected, removed the s in DESs

Page 4, line 95: There seems to be one space to many.

REPLY - corrected

Page 4, line 102: it should be of instead of in

REPLY - corrected

Page 4, line 106: missing space ...infarction(MI)....

REPLY - corrected

Page 5, line 117: there seems to be a 0 missing. please correct: (P<.10)

REPLY – thank you for the  observation - corrected

Page 5, line 128: There seems to be one space to many.

REPLY - corrected

Page 5, line 136/137: please write consistently ; There seems to be one space to many.

REPLY – Thanks for spotting this on. We have amended the text accordingly.

Page 14, line 203: ...with EES or R-ZES drug-eluting stents followed up to five years...

REPLY - ?

Reviewer 3 Report

The authors significantly improved and clarified their manuscript.

Author Response

We want to thank the reviewer for his comments which haave contributed to improve our manuscript.

Reviewer 4 Report

In the “limitations of the study”, the use of propensity score doesn’t generate randomized study. This sentence must be rewritten with more moderation

OK this sentences has be rewritten with more moderation

The primary endpoints included hospitalizations is not describied. What kind of hospitalisation ? No description in the methods and not detailed in the results ( stroke CHF others )

Line 107 hospitalizations are not still describied ++++ with no definition

Line 139 HF hospitalisation is describied in the flow chart but not in the primary endpoints

There is no flow chart ? How many patients was excluded ? Why ? the reasons ?

Ok for the Flow Chart but in Line 139 a space misses between Flow and Chart (Flow Chart)

2406 patients were excluded ? Why ? how many lost of follow up ? how many without demographic ou angiographic information ? What kind of information were missing ?

What is the mean follow-up in each group ?

No information in table 1

In table 1 how many patient in EES group ? in ZES group ?

OK number patients of each group is describied

Radial access is different in each group ? why ? can it explain the difference in the 2 groups ? explain in the discussion ? 2

No explanation is noticed in the discussion. It can be modified the resulted and modified the pronostic of PTCA.

The authors must describied the population in two tables: First demographic and reason for admission, Second angiographic and revascularization characteristic

the authors did not take this remark into account. I think it is essential for the clarity of this study. Table 1 for demographic data, table 2 for angiographic data

Mortality must be divided en total mortality and cardiovascular mortality in each group with two different curves.

the authors did not take this remark into account. I think it is essential for the clarity of this study.

Repeat PCI must be describied as stent thrombosis (ARC definition) emergent PCI, ischaemia driven PCI in each group

the authors did not take this remark into account. I think it is essential for the clarity of this study.

In each group, what kind of treatment ? what kind of platelet agents ? how long was the DAPT  ( mean, mediane and extreme) ? These results must be explain in the discussion.

the authors did not take this remark into account. I think it is essential for the clarity of this study.

I confirm that the authors cannot answer the question without providing these essential clarification.

Author Response

Reviewer 4

We thank the reviewer for detailed evaluation of our paper which is now improved substantially. We feel that we have addressed all his comments.

In the “limitations of the study”, the use of propensity score doesn’t generate randomized study. This sentence must be rewritten with more moderation

REPLY. We have rephrased the sentence playing down these statements:”The use of propensity score generated a balanced population mitigating confounding. However, residual confounding could not be fully excluded. Consistency of the results in all prespecified subgroups support the reliability of the findings”.

The primary endpoints included hospitalizations is not described. What kind of hospitalisation ? No description in the methods and not detailed in the results ( stroke CHF others )

Line 107 hospitalizations are not still described ++++ with no definition

REPLY – hospitalizations consisted of heart failure hospitalizations. There were actually a small number  hospitalizations for HF in both groups with limited information on the timing of occurrence. In view of the few occurrences and the limited  available data which prevent to present individual results for this outcome and to adhere to more stringent definition of ischemic MACCEs in line with current stent literature we now do not include hospitalizations in the final MACCE endpoint. This intervention does not change the outcomes.

Line 139 HF hospitalisation is described in the flow chart but not in the primary endpoints

There is no flow chart ? How many patients was excluded ? Why ? the reasons ?

Ok for the Flow Chart but in Line 139 a space misses between Flow and Chart (Flow Chart)

REPLY – Thank you. We have now corrected this.

2406 patients were excluded ? Why ? how many lost of follow up ? how many without demographic ou angiographic information ? What kind of information were missing ?

REPLY -   Patients were excluded if there was incomplete information on variables of interest or outcome data at long-term follow-up were not fully reported. We have now added this information to the text and amended accordingly the flow chart figure.  

What is the mean follow-up in each group ?

REPLY -  Mean follow up for EES was 1686 days vs 1652 for R-ZES including all patients

No information in table 1

In table 1 how many patient in EES group ? in ZES group ?

OK number patients of each group is described

Radial access is different in each group ? why ? can it explain the difference in the 2 groups ? explain in the discussion ? 2

No explanation is noticed in the discussion. It can be modified the resulted and modified the prognostic of PTCA.

REPLY – We already reported in the Table 1 the percentage of radial access in both  groups which was comparable: EES 88% vs R-ZES 90%. There was an error reporting the P value of 0.03. The precise P value is 0.3.

The authors must describe the population in two tables: First demographic and reason for admission, Second angiographic and revascularization characteristic

the authors did not take this remark into account. I think it is essential for the clarity of this study. Table 1 for demographic data, table 2 for angiographic data

REPLY. We have added one more figure following reviewer suggestions and the total number of figures and tables is expanded. We do not feel we should split the baseline and procedural characteristics of patients into further sub-tables which would be rather confusing to the reader. We hardly recall such a splitting operation in other similar studies published which instead presented the overall table of baseline and procedural characteristics as we did.

Mortality must be divided en total mortality and cardiovascular mortality in each group with two different curves.

the authors did not take this remark into account. I think it is essential for the clarity of this study.

REPLY. We have already stated  in the first revision that all-cause mortality was the prespecified endpoint. No information on the mortality subtype is available. All-cause mortality is a an established  primary endpoint of major clinical trials in the field.

Repeat PCI must be describied as stent thrombosis (ARC definition) emergent PCI, ischaemia driven PCI in each group

the authors did not take this remark into account. I think it is essential for the clarity of this study.

REPLY. As already stated in the previous rebuttal, repeat PCI mostly driven by restenoses occurrences is NOT stent thrombosis and should not incorporate it,  which is rather a cause for myocardial infarction and is always presented as a separate endpoint. There were indeed no cases of stent thrombosis among the repeat PCI events. One may refer to “Garcia - Standardized End Point Definitions for Coronary Intervention Trials: The Academic Research Consortium-2 Consensus Document. European Heart Journal. 2018;39:2192-207” for a closer look to the definition of repeat PCI.

In each group, what kind of treatment ? what kind of platelet agents ? how long was the DAPT  ( mean, mediane and extreme) ? These results must be explain in the discussion.

the authors did not take this remark into account. I think it is essential for the clarity of this study.

REPLY  In the text we  have already discussed this point that dual antiplatlet therapy with aspirin 75-100 mg  and a P2Y12 inhibitor was mandated in all patients  for at least one year.  No details on P2Y12 subtypes are available and could not be added.

I confirm that the authors cannot answer the question without providing these essential clarification.

Round 3

Reviewer 4 Report

REPLY  In the text we  have already discussed this point that dual antiplatlet therapy with aspirin 75-100 mg  and a P2Y12 inhibitor was mandated in all patients  for at least one year.  No details on P2Y12 subtypes are available and could not be Added. 

This question is fundamental . Treatment with cloidogrel is not the same as ticgarelor or prasugrel. If there some imbalance beetween the two groups regarding this kind of treatment , the results is not valided .

The question of DAPT duration is fundamental  I can't belive that all  et every  patient are on DAPT. Even in tne best managed studies , the mean duration fluctuates between 8 or 9 months and 13 or 15 months . These data are fundamental  

 "As already stated in the previous rebuttal, repeat PCI mostly driven by restenoses occurrences". I agree with the authors that most of repeat PTCA is restenosis but not all . If there some imbalance beetween the two groups regarding this kind of ECV , the results is not valided . for example , more PTCA per patients more resténsosis

"We have added one more figure following reviewer suggestions and the total number of figures and tables is expanded."

If the figures and the tables are expanded , the authors could  dropped the data in the annexes accessible via internet 

"We have already stated  in the first revision that all-cause mortality was the prespecified endpoint. No information on the mortality subtype is available. All-cause mortality is a an established  primary endpoint of major clinical trials in the field"

The primary endpoint has to be cardio-vascular morbi-mortality with in a subtype all cause  mortality. CV mortality is a relation with the aim of the study (efficacy) et total mortality the safety . Most of drugs decrease  cardiovascular mortality but increase  total mortality .  It is important to clarify these two situation .In this  study the endopoint is  to imprecise. We want to know waht is the effect of these stents: decrease cardiovascular mortality and decresase all -cause mortality, or increase cv mortalit with decrease total mortality .